# Role of Clathrin Light Chains in Regulating Invadopodia Formation

**DOI:** 10.3390/cells10020451

**Published:** 2021-02-20

**Authors:** Markus Mukenhirn, Francesco Muraca, Delia Bucher, Edgar Asberger, Elisa Cappio Barazzone, Elisabetta Ada Cavalcanti-Adam, Steeve Boulant

**Affiliations:** 1Department of Infectious Diseases, Virology, University Hospital Heidelberg, Im Neuenheimer Feld 344, 69120 Heidelberg, Germany; mukenhirn@gmail.com (M.M.); francesco.muraca@uni-heidelberg.de (F.M.); delibucher@gmail.com (D.B.); edgarr@freenet.de (E.A.); elisa.cappio@gmail.com (E.C.B.); 2Max Planck Institute for Medical Research, 69120 Heidelberg, Germany; eacavalcanti@mr.mpg.de; 3German Cancer Research Center (DKFZ), 69120 Heidelberg, Germany

**Keywords:** clathrin light chain, invadopodia, MMP14: actin, membrane trafficking

## Abstract

One of the most fundamental processes of the cell is the uptake of molecules from the surrounding environment. Clathrin-mediated endocytosis (CME) is the best-described uptake pathway and regulates nutrient uptake, protein and lipid turnover at the plasma membrane (PM), cell signaling, cell motility and cell polarity. The main protein in CME is clathrin, which assembles as a triskelion-looking building block made of three clathrin heavy chains and three clathrin light chains. Compared to clathrin heavy chains (CHCs), the role of the two isoforms of clathrin light chains (CLCA and CLCB) is poorly understood. Here, we confirm that the simultaneous deletion of both CLCA/B causes abnormal actin structures at the ventral PM and we describe them, for the first time, as functional invadopodia rather than disorganized actin-cytoskeleton assembly sites. Their identification is based on the occurrence of common invadopodia markers as well as functional invadopodia activity characterized by an increased local proteolytic activity of the extracellular matrix proteins. We demonstrate that CLCA/B deletion impacts the intracellular trafficking and recovery of the matrix metalloproteinase 14 (MMP14) leading to its accumulation at the plasma membrane and induction of invadopodia formation. Importantly, we show that invadopodia formation can be prevented by depletion of MMP14. As such, we propose that CLCA/B regulate invadopodia formation by regulating MMP14 delivery to the plasma membrane.

## 1. Introduction

Uptake of biomolecules from the extracellular environment is the most fundamental cellular process. Clathrin-mediated endocytosis (CME) is the most important and best characterized endocytic pathway and regulates nutrient uptake, protein and lipid turnover at the plasma membrane, cell signaling, cell motility and cellular polarity [1,2,3]. CME is a very complex process that relies on the coordinated recruitment of multiple proteins to initiate the formation of a clathrin-coated pit (CCP) at the plasma membrane, which will drive membrane invagination and ultimately lead to the release of clathrin-coated vesicles (CCVs) into the cytosol [4,5]. The building blocks for CCP and CCV formation are clathrin triskelia. A clathrin triskelion consists of three clathrin heavy chains (CHCs) and three clathrin light chains (CLCs) [3]. CHCs are fundamental for CME as they form the structural backbone of the triskelia while CLCs appear to usually be dispensable for CME [6,7].

Vertebrates possess two clathrin light chain isoforms, clathrin light chain A (CLCA) and clathrin light chain B (CLCB), which are the result of the duplication of an ancient gene [8]. CLCA and CLCB are ubiquitously expressed in mammalian tissue, with the exception of lymphoid cells that seem to have only CLCA [9]. CLCA and CLCB are similarly organized and have about 60% sequence identity. CLCB has a unique serine phosphorylation site close to the N terminus of the protein; whereas, CLCA has an additional Hsc70 stimulating region [2,10]. The functions of CLCA and CLCB remain poorly understood. Silencing experiments of both CLCA and CLCB reported little to no impact on CME of transferrin or epidermal growth factor (EGF) [6,7]. However, recent work exploiting CRISPR/Cas9 editing technology shows that cells expressing only CLCB exhibit an increased rate of CME compared to cells expressing only CLCA or cells expressing both CLCs [11]. CLCs were also reported to be important for the uptake of a subset of G protein-coupled receptors (GPCRs) [9,12] and, more recently, it has shown that site specific phosphorylation of CLCB regulates GPCR endocytosis but not endocytosis of transferrin, by promoting the transition from flat to curve clathrin lattices [13]. Transition from flat to curved lattices represents a key step during CME [14] and in vitro studies have proposed that CLCs contribute to the tensile strength of the clathrin coat, which will help bending artificial membranes [15]. CLCs were reported to interact with the actin cytoskeleton via binding to huntingtin interacting protein 1 (Hip1R) [2,16]. Coupling of CLCs to the actin-cytoskeleton has been shown to be important for endocytosis of unconventionally large cargo such as bacteria or certain viruses [17,18] or at sites of increased membrane tension [19]. Furthermore, CLCA acts together with Myosin VI to generate the force required for endocytosis at polarized tissue [20]. As such, there is growing evidence for a light chain specific function in regulating CME.

Beside their role in endocytosis, CLCs have been described to have an important function in intracellular trafficking and cargo recycling to/from the plasma membrane. Gene silencing of both CLCA and CLCB expression results in a perturbed trafficking of the cation-independent mannose-6-phosphate receptor which was found to be trapped in the trans-Golgi compartment [7]. Interestingly, knock-out of both CLCA and CLCB was also found to induce the accumulation of actin patches but their identity and function remains unclear [7,21,22]. Similarly, CLCs were shown to be important for the recycling of β1 integrin back to the plasma membrane [21] and, thereby, influencing cell migration [21] and cell spreading [11,23]. Finally, deletion of CLCA in B cells impairs both their maturation in the germinal center and antibody isotype switching. Selective recycling and internalization of signaling receptors like TGFβR2, CXCR4, and DOR mediated by CLCA were proposed to be the underlying mechanisms [9]. 

In this study, we use CRISPR/Cas9 mediated genome editing to sequentially deplete both CLCs. We confirm the appearance of aberrant actin accumulations following both CLC isoforms deletion. Using qualitative and functional assays, we describe these actin structures as functional invadopodia based on the presence of common invadopodia markers as well as their association with discrete proteolytic activity of the extracellular matrix. We could show that deletion of both CLCs induces the accumulation of the matrix metalloproteinase MMP14 at the plasma membrane, which, in turn, promotes invadopodia formation. Accumulation of MMP14 was due to its increased trafficking from intracellular compartments to the plasma membrane and not to a defect in internalization. As such, our work further highlights the importance of CLCs in regulating intracellular trafficking of cargo molecules and describes a novel function of CLCs in regulating invadopodia formation.

## 2. Materials and Methods

### 2.1. Cell lines and Cell Culture

U373 and HEK293T cells were maintained in DMEM (Thermo Fisher, Waltham, MA, USA, #41965-039) and in IMDM (Thermo Fisher, #21980-032) media, respectively. Both cell culture media were supplemented with 10% FBS (Biochrom, S0615) and penicillin/streptomycin (Thermo Fisher, 15140). Cells were grown at 37 °C in the presence of 5% CO2. U373 cells stably expressing AP2-eGFP were created by transfection of a plasmid expressing the sigma2 subunit of AP2 fused to eGFP (Ehrlich et al., 2004). Selection of stable KO cell lines was performed using a final concentration of 500 µg/mL Geneticin/G418 (Thermo Fisher, 10131-027), 2 µg/mL Puromycin (Merck-Sigma Aldrich, St. Louis, MO, USA, P9620) and/or 10 µg/mL Blasticidin (Thermo Fisher, R21001).

### 2.2. Antibodies

Mouse monoclonal antibody against clathrin light chains (Con.1) (Merck-Sigma Aldrich, C1985) was used at 1:1000 for immunofluorescence (IF) and (1:200) Western blotting (WB). Mouse monoclonal antibody against clathrin heavy chains (TD.1) (Abcam, Cambridge, UK, ab24578) was used 1:1000 for WB. Mouse monoclonal antibody against Arp3 (Merck-Sigma Aldrich, A-5979) was used at 1:1000 for IF. Rabbit polyclonal anti-MMP14 antibody (Abcam, ab51074) was used at 1:1000 for IF and WB and monoclonal antibody against β actin (A5441); used 1:5000 for WB). Rabbit polyclonal anti-HIP1R (Merck-Millipore, Burlington, MA, USA, AB9882) was used 1:100 for IF. Mouse monoclonal antibody against Cortactin (p80/85) (Merck-Millipore, 05-180) was used at 1:1000 for IF. Phalloidin coupled with Alexa fluorophore 647 (Thermo Fisher, A-22287) was used at 1:100 for IF. Secondary anti-mouse antibodies coupled to Horseradish Peroxidase (HRP) (GE Healthcare, Chicago, IL, USA, NA931) or anti-rabbit IgG HRP (GE Healthcare, NA934) was used 1:50000 for WB. Secondary anti-mouse antibodies coupled to Alexa Fluorophore 568 (Thermo Fisher, A-11004) and anti-rabbit antibodies coupled to Alexa Fluorophore 568 (Thermo Fisher, A-11011) were used (1:1000) for IF.

### 2.3. Plasmids, Viral Vectors and gRNA

The lentiviral backbones lentiCRISPR v2 (#52961), lentiCRISPR v2-Blast (#83480) and plko.1 neo (13425) as well as helper plasmids pMDG.2 (#12259) and psPAX (#12260) were purchased from Addgene. MMP14 pHluorin plasmid was kindly provided by Kay Oliver Schink, Harald Stenmark and Philippe Chavrier. The BacMam mCherry-zyxin viral vector was used for transient zyxin expression [24]. CLCB QQN RFP vector was provided by P. McPherson. The Gateway Destination vector pDest RFP-C was used to clone CLCA and CLCB at the C-terminal end of RFP as well as the cDNA of human CLCA and CLCB (pENTR CLCA and pENTR CLCB) were obtained from the Vector and Clone Repository and repository genome wide cDNA library Gateway Full ORF Clones distributed by the GPCF available at the DKFZ. The following target gRNA sequences were used for genome editing: For single KO clones: CLCA−− (5’GCCTGGACGGCGGCGCCCCC3`) and CLCB−/− (5’AGAGAACGACGAGGGCTTCG3´). For double KO clones: CLCA−/− (5´GACGCGCCCGCACTCTCACC 3)´ and CLCB−/− (5´AGAGAACGACGAGGGCTTCG 3´). Following target sequence was used for shRNA mediated protein knock-down: MMP14 (5’CATTGCATCTTCCCTAGATAG 3´).

### 2.4. Western Blot (WB)

Protein samples for Western blot analysis (WB) were produced from cell lysates, cells were detached and washed twice with PBS and lysed for 30 min on ice by resuspending them in appropriate volume of RIPA buffer (150 mM sodium chloride, 1.0% (*v*/*v*) NP-40, 0.5% sodium deoxycholate, 0.1% sodium dodecyl sulphate (SDS) and 50 mM Tris, pH 8.0) containing the complete protease inhibitor cocktail (Roche, #118735800001)). Protein concentration was measured using the Protein assay Kit II (BioRad, Hercules, CA, USA, #500112). Sodium dodecyl sulphate polyacrylamide gel electrophoresis (SDS-PAGE) gels were prepared with appropriate concentrations of the separating gel. Protein samples were mixed with 4x Laemmli buffer (0.2 M Tris-HCl pH 6.8, 0.05 M EDTA,40% 8*v*/*v*) glycerol, 8% 8*w*/*v*) SDS, 4% 8% (*w*/*v*) SDS, 4% (*v*/*v*) β-mercaptoethanol, 0.03% (*w*/*v*) bromophenol blue), heated for 10 min at 95 °C. 5 µL Precision Plus Protein Dual Color Standards marker (BioRad, 1610374) was used together with the protein samples in SDS-Tris-Glycine buffer (25 mM Tris-base, 200 mM Glycine, 1% (*w*/*v*) SDS). SDS-PAGE was performed at 80 V until samples reached the running gel and the voltage was changed to 120 V until desired separation was achieved. Proteins were transferred to a nitrocellulose blotting membrane (0.45 µm; GE Healthcare, #10600003) by wet blot transfer at 100 V for 80 min, nitrocellulose membrane were blocked with 5% milk in TBST (50 mM Tris-HCL pH 7.5, 150 mM NaCl, 0.1% Tween) for 1 h at room temperature. Incubation with primary antibody diluted in TBST with 5% milk was performed overnight at 4 °C. After four washes for 5 min with TBST, membranes were incubated with secondary antibody diluted in TBST with 5% milk for 1 h at room temperature. The membranes were washed four times for 5 min with TBST and developed using the ECL Western Blotting detection reagents (GE Healthcare, RNP2106) or Western Bright Chemiluminescence Substrate Sirius (Biozym, 541021) using high performance chemiluminescence films (GE Healthcare, #28906837).

### 2.5. Cloning of Guide RNA (gRNA) or shRNA Encoding Vector for Lentivirus Production

Forward and reverse oligonucleotides containing guide RNA (gRNA) sequence were designed based on the open source gRNA design tools (https://portals.broadinstitute.org/gpp/public/analysis-tools/sgrna-design (accessed on 15 July 2015) and http://chopchop.cbu.uib.no/) (accessed on 15 July 2015). Forward and reverse oligonucleotides contaning shRNA sequence were designed according to RNA interference (RNAi) consortium (TRC) (https://www.broadinstitute.org/scientific-community/science/projects/rnai-consortium/rnai-consortium) (accessed on 17 August 2020). All oligonucleotides were ordered from Eurofins. Complementary oligonucleotides were designed to generate 5′ and 3′ overhangs after annealing complementary to BSMBI restriction sites for gRNA or AgeI and EcoRI restriction sites for shRNA, respectively, for direct ligation into linearized vectors. Oligonucleotides were diluted in H_2_O at a concentration of 1 µg/µL. 2.5 µL of forward and reverse oligonucleotides were mixed with 5 µL NEB Buffer 2 (New England Biolabs, Ipswich, MA, USA, B7002S) and 40 µL H_2_O. The oligonucleotide mix was heated at 95 °C for 5 min and slowly cooled down to room temperature for annealing of both strands. For the ligation reaction, 150 ng of lentiCRISPR v2 or lentiCRISPR v2-Blast backbone for gRNA cloning or 150 ng of plko.1 neo backbone for shRNA, which has been digested with either BSMBI (R0580) or AgeI-HF (R3552S) and EcoRI-HF (R3101S) purchased from NEB, and 2 µL annealed oligonucleotides were used. A 20 µL ligation reaction was set up using T4 Ligase (New England Biolabs, M0202S) and incubated for 1 h at room temperature. The whole ligation mix was used for transformation into DH5α (Thermo Fisher, 18265017). Positive colonies were confirmed by sequencing.

### 2.6. Gateway Cloning

To generate expression vectors, Gateway LR Clonase II Enzyme Mix (Thermo Fisher, 11791) was used. In brief, LR-recombination reactions between Gateway entry vectors containing attL-sites and Gateway destination plasmids containing attR-sites were set up by mixing 150 ng of the entry vector with 150 ng of the destination vector. TE buffer was added to reach a final volume of 8 µL. Then, 2 µL of LR Clonase II Enzyme Mix were added and incubated for 1–3 h at 25 °C. 1 µL of Proteinase K was added to the reaction mix and incubated for 10 min at 37 °C. The reaction mix was transformed into DH5α.

### 2.7. Lentivirus Production

For lentivirus production, Hek293T cells were cultured in 10 cm dishes until 80% confluence. The cells were transfected with pMDG.2, psPAX, and the shRNA encoding pLKO.1 vector or the pWPI vector containing the gene of interest using PEI. 50 µL of PEI (1 mg/mL) was diluted in 200 µL Opti-MEM (Thermo Fisher, 31985062) and mixed well. An amount of 4 µg pMDG.2, 4 µg psPAX, and 8 µg pLKO.1 or pWPI were mixed with Opti-MEM to a final volume of 250 µL and mixed well. After 5 min, both solutions were mixed, vortexed and incubated for 20 min at room temperature. Then the transfection mix was added dropwise to cells. Culture medium was exchanged the next day. After another two days, the supernatant was harvested, centrifuged (10 min, 4000× *g*) and filtered through a syringe filter (Millex-HA, 0.45 µm, Merck-Millipore, SLHA033SS). For short time storage, the lentivirus containing supernatant was kept at 4 °C; for long time storage, the supernatant was aliquoted and stored at −80 °C.

### 2.8. Transfection and Viral Transduction

Transfection of cells was done using Lipofectamine 3000 (Thermo Fisher, L300008) if not stated otherwise. Cells were plated in 6-well plates one day before transfection. The next day, cells were transfected at 70–80% confluence. 2.5 µg DNA together with 10µL of P3000 Reagent and 7.5 µL Lipofectamine 3000 were separately mixed with 125 µL Opti-MEM. The two solutions were mixed together. After incubation for 15 min at room temperature, the transfection mix was added dropwise onto the cells. For live-cell imaging of cells transiently expressing fluorescently tagged focal adhesion (FA) proteins, the growth medium was exchanged for fresh growth medium after 6 h. The transfected cells were seeded 24 h after transfection and imaged 6–8 h after seeding.

For transient expression of mCherry-zyxin, cells were transduced with BacMam mCherry-zyxin. Cells were seeded with BacMam mCherry-zyxin in appropriate dishes and were used for experiments 1 day after seeding. 3.5 µL of the P3 stock was used per 10,000 cells. For lentiviral transduction for generation of stable cell lines, 10,000 cells per well were seeded into 6-well plates together with 500 µL of lentivirus containing supernatant. After 2 to 3 days, growth medium was exchanged to selection medium with appropriate antibiotics. Cells were sub-cultured in selection medium for 2 weeks and then used for experiments. 

### 2.9. Wide Field Epifluorescence Microscopy

Widefield epifluorescence microscopy was performed with an inverted Nikon Ti microscope with a 20× (0.75 numerical aperture, Plan Apo λ, Nikon Instruments, Amstelveen, Netherlands) “air” objective or a 40× (1.3 numerical aperture, Plan Fluor, Nikon Instruments) oil immersion objective. The microscope is equipped with a digital camera (DS-Qi1MC, Nikon Instruments) and a LED fluorescence source LED (Lumencor-Sola).

### 2.10. Total Internal Reflection Fluorescence (TIRF) Microscopy

TIRF microscopy was performed with an inverted Ti microscope (Nikon Instruments) with objective TIRF illumination, with a 60× (1.49 numerical aperture, Apo TIRF, Nikon Instruments) oil immersion objective and an EMCCD camera (Andor iXon Ultra DU-897U) and an inverted Axiovert200M (Zeiss) Orbital-TIRF system, using an alpha Plan-Apochromat 100×/1,46 Oil DIC oil immersion objective and an Hamamatsu EMCCD 9100-50 camera.

### 2.11. Spinning Disc Confocal Microscopy

Confocal live fluorescence cell imaging was performed with an inverted spinning disc confocal microscope (Nikon Instruments, PerkinElmer) equipped with 60× (1.42 numerical aperture, Apo TIRF, Nikon) oil immersion objective and a CMOS camera (Hamamatsu Ocra Flash 4) or an EMCCD camera (Hamamatsu C9100-23B). The biological samples were kept in an environment control chamber at 37 °C with 5% CO2. 

### 2.12. Fluorescence Recovery after Photobleaching (FRAP)

MMP14-pHluorin was imaged and photobleached using a 488-nm laser light. A defined region of interest (ROI) was photobleached at high laser power. Live-cell confocal imaging was performed for a total of 25 min with at least 3 min and 30 s pre-bleach phase and 20 min recovery phase and at a frame rate of 5 s per frame.The fluorescence recovery in ROI was normalized and analyzed using the easyfrab-web tool: https://easyfrap.vmnet.upatras.gr/ (accessed on 6 April 2020).

### 2.13. Immunofluorescence

Cells growing on glass coverslips (#1.5, diameter 12 mm, purchased from Thermo Scientific or 24 mm purchased from Marienfeld) were washed twice with PBS before being fixed with 2% PFA in PBS for 20 min at room temperature or overnight at 4 °C. All following steps were performed at room temperature. After three washes with PBS, cells were permeabilized with 0.5% TritonX in PBS for 15 min followed by blocking with 1% BSA in PBS for 30 min. Samples were incubated with appropriate primary antibody diluted in 1% BSA in PBS for 1 h. After three washes with PBS, samples were stained with appropriate secondary antibody or Phalloidin labeled with Alexa Fluor 647 for 45 min. After four washes with PBS, samples were quickly rinsed H_2_O and mounted with ProLong Gold antifade containing DAPI (Thermo Fisher, P36931).

### 2.14. ECM Degradation Assay

ECM degradation assay was performed as published before [25]. In short, a solution of 0.2% gelatin (from porcine skin; Sigma, G-2500) was prepared in PBS. To dissolve the gelatin, the solution was heated up to 37 °C for 30 min. For sterilization, the solution was filtered using a syringe filter membrane (Millex-GS, 0.22 µm, Merck-Millipore, SLGS033SB). The gelatin was labelled with Alexa Fluor 647 NHS Ester solution (10 mg/mL, Thermo Fisher, A37573) for one hour at room temperature. Labelled gelatin was dialyzed against PBS first for 2 h at room temperature then overnight at 4 °C using Slide-A-Lyzer MINI Dialysis Device (20kMWCO, 0.5 mL; Thermo Scientific, #88402) to remove the free dye. 

Coverslips for cell seeding were first coated using 50 µg/mL of poly-L-lysine (PLL Sigma, P8920) for 20 min at room temperature. Coverslips were washed three times with PBS and then fixed with 0.5% glutaraldehyde (Merck-Sigma Aldrich, G5882) in PBS for 15 min at room temperature. Following extensive washes with PBS, solution of Alexa Fluor 647 labelled gelatin and unlabeled gelatin (1:8 ratio) was preheated to 37 °C for 30 min and used to coat the PLL-coated coverslips. Coverslips were inverted on a drop of gelatin mix (80 µL for 12 mm diameter coverslips; 120 µL for 25 mm diameter coverslips) and incubated for 10 min at room temperature. Coverslips were washed three times with PBS. The coated coverslips were directly used for experiments.

### 2.15. Microscopy-Based Internalization Assay of MMP14

Antibody based internalization assay was performed using a similar protocol as previously described (Majeed et al., 2014). In short, cells were plated on glass coverslips overnight. Following serum starvation for 1 h, cells were labelled with MMP14 antibody (1:100) on ice for 15 min. Cells were then washed 3 times with ice cold PBS to remove unbound antibodies. To determine “total surface MMP14” levels, cells were directly fixed using 2% PFA in PBS and immunostained using the anti-rabbit secondary antibody (without permeabilization). To determine “internal MMP14” levels, following incubation of the cells with MMP14 antibody (1:100) on ice for 15 min, cells were incubated at 37 °C for 30 min in order to start MMP14 internalization. For the measurement of “Internalized MMP14” levels, following incubation at 37 °C for 30 min, cell surface was stripped using ice-cold PBS at pH 2.5, to removed non-internalized primary antibodies, followed by fixation with 2% PFA in PBS and immunostaining using the anti-rabbit secondary antibody (with permeabilization). Imaging was performed using confocal microscopy. Z-stack series were acquired every 0.5 µm for a total of 7.5 µm. Fiji was used to quantify MMP14 levels. Following background subtraction, mean fluorescent intensity was measured for individual cells.

### 2.16. Transferrin Uptake

Cells were incubated with 50 µg/mL Alexa Fluor 647 labelled Transferrin (Invitrogen, T-23366) for 5 min at 37 °C. Unbound transferrin was removed by washing 3 times with ice-cold PBS and cell surface bound transferrin was washed off using ice-cold PBS at pH 2.5 (acid wash). Following fixation of cells with 2% PFA, internalized transferrin was visualized using confocal microscopy and quantified using Fiji.

### 2.17. Flow Cytometry-Based Internalization Assay

Internalization assay was, in principle, performed as described before (Pujol et al., 2016 and Figure 5A). Cells were dissociated from cell culture flasks using StemPro AccutaseTM (Thermo Fisher, A1110501) and resuspended in serum free culture medium and kept on ice. Cells were incubated with saturating concentration of MMP14 antibody (1:100) on ice for 40 min, and then washed with cold serum free culture medium before being shifted to 37 °C for the respective time intervals between 0 and 40 min. Following MMP14 internalization, cells were incubated with anti-rabbit Alexa Fluor 647 (1:10000) for 30 min, washed and analyzed by flow cytometry (FACS Canto II, BD Biosciences). Data was analyzed using FlowJoTM version 10.

### 2.18. Flow Cytometry-Based MMP14 Surface Detection

Similar to the Internalization assay, cells were dissociated from cell culture flasks using StemPro AccutaseTM (Life Technologies, A1110501) and resuspended in serum free culture medium, kept on ice. Cells were incubated with rabbit anti-MMP14 antibodies for 30 min followed by fixation and permeabilization with Cytofix/CytopermTM fixation/Permeabilization Kit (BD Biosciences, 554714). Cells were then incubated with anti-rabbit Alexa Fluor 647 for 30 min and analyzed by flow cytometry (FACS Canto II, BD Biosciences). Data was analyzed using FlowJoTM version 10.

### 2.19. Surface Biotinylation Assay

A total of 5 × 10^5^ U373 cells (WT and CLC dKO) were seeded in T25 flasks and left to adhere overnight. Afterwards, cells were put on ice and washed three times with cold PBS supplemented with 1mM CaCl2 and 0.5 mM MgCl2 (PBS-Ca/Mg). Cells were then treated with 1mg/mL Ez-Link© Sulfo-NHS-SS-biotin (Thermo Fisher) in PBS-Ca/Mg on ice for 15 min, followed by two 5 min incubations with cold 100mM glycine in PBS-Ca/Mg to quench free biotin. A total of three washes with PBS-Ca/Mg were performed and cells were incubated for 30 min at 37 °C to allow internalization of MMP14. Following the incubation, cells were put back on ice and surface biotin was stripped by three 10 min incubations with 50 mM L-glutathione (Merck-Sigma Aldrich), in solution with 75 mM NaCl, and 10 mM EDTA (pH 7.5). Free –SH groups were then alkylated by three successive incubations with 5mg/mL Iodoacetamide (IAA) in PBS-Ca/Mg. To allow for the recycling of internalized MMP14, cells were put back at 37 °C for an additional 30 min. After this short incubation the recycled surface biotin was again stripped as described above. Cells were lysed using a buffer composed of 1% Triton X-100, 0.1% SDS, 150 mM NaCl, 5 mM EDTA and 50 mM HEPES at pH 7.5, and incubated on ice for 30 min, followed by short sonication and centrifugation to remove cell debris. Protein concentration was assessed and normalized using a DC protein assay kit (Biorad) and then the same amount of protein per sample was added to high capacity NeutrAvidin agarose beads (Thermo Fisher), and incubated overnight at 4 °C under gentle agitation. The following day, the supernatant was removed and beads were washed two times with 1M NaCl, 50mM HEPES, 0.1% Triton X-100 (pH 7.4) and two times with 50mM HEPES (pH 7.4), followed by a 20 min incubation at RT with Laemmli buffer containing 5% SDS, 100 mM NaCl, 100 mM DTT, and 5% B-Mercaptoethanol. The eluted biotinylated proteins were then boiled and loaded onto acrylamide gels for Western blotting.

### 2.20. Inhibition of Protein Synthesis by Cycloheximide

Cells were seeded on coverslips (IFA) or appropriate cell culture dishes 16h prior experiment. Cells were incubated for 1, 6 and 12 h with 10µg/mL cycloheximide (Sigma). For surface MMP14 samples, cells were incubated on ice with MMP14 antibody for 30 min prior fixation with 2% PFA/PBS. Total MMP14 samples were either lysed with Laemmli buffer or fixed with 2% PFA/PBS and processed for WB analysis or IFA, respectively.

### 2.21. Quantification of Actin Patches

Confocal microscopy images of cells stained with actin marker Alexa Fluor 647-labelled Phalloidin were analyzed by eye whether cells show aberrant actin patches in addition to normal actin filaments.

### 2.22. Tracking of Clathrin Structures

The software ilastik (http://ilastik.org (accessed on 17 August 2020)) was used for tracking CME events as described in [24], in brief: Images were segmented using pixel classification and object classification workflows. The automatic tracking workflow was used for tracking single clathrin events. The maximum distance of two neighboring objects was set to 5 to avoid merging. Tracking results were analyzed using an automated KNIME workflow (lifetime, maximal fluorescence intensity and average position).

## 3. Results

### 3.1. Loss of Both CLCA and CLCB Induces Aberrant Actin Patches That Can Be Rescued by Expression of a Single CLC

To investigate the functions of CLCA and CLCB, we used CRISPR/Cas9 genome editing to sequentially deplete both CLC isoforms in the astrocyte derived cell line U373. Following selection and single cell cloning, successful genome editing was confirmed by immunoblotting and sequencing. In WT cells, both CLCA and CLCB were detectable whereas only CLCB and CLCA were detectable in CLCA−/− and CLCB−/− cells, respectively (Figure 1A). Knock-out of a single clathrin light chain was associated with a 20% increase in the expression level of the other isoform (Figure 1A). Deletion of both CLCA and CLCB in the double knock-out U373 cell line was confirmed both by immunoblotting (Figure 1A) and by immunofluorescence assay (IFA) (Figure 1B) and CLC deletion had minimal effect on CHC expression (Supplementary Figure 1A). To monitor the impact of CLC deletion on CME, we performed a transferrin internalization assay in both single (CLCA−/− and CLCB−/−) and double CLC KO cell lines and compared it to WT cells. As expected, knock-out of either or both CLCs did not impact the clathrin-mediated internalization of transferrin (Appendix A). Complementarily, we monitored the dynamic of CME in WT cells and compared it to cells lacking CLCs. Analysis of the lifetime of the CCPs present at the ventral membrane of cells showed no significant differences between WT and CLCs depleted cells (Appendix A). Together, these results confirm previous observations that CLCs do not participate in regulating CME of “conventional” cargo molecules [6,7].

It has been previously observed upon siRNA-mediated CLCA/B knock-down that actin patches assemble at the ventral plasma membrane of cells [7,21]. Here, upon CRISPR/Cas9 mediated knock-out of CLCA/B, we report a similar accumulation of actin at the plasma membrane. In WT cells, actin was mostly forming fibrillar structures corresponding to stress fibers (Figure 1C,D). On the contrary, upon deletion of both isoforms of CLCs, actin was found in patches clustered under the nucleus (Figure 1C,D). Importantly, in cells depleted of either CLCA or CLCB, actin was mostly found in fibrillar structures and cells did not display actin patches (Figure 1D and Appendix A). This observation strongly suggests that both CLCs share redundant functions in regulating formation of these actin patches. Similar results were observed when using different clones of CLCA and/or CLCB knock-out cell lines strongly supporting that this phenotype is specific for the deletion of CLCs and not a consequence of single cell cloning (data not shown). 

To validate the critical role of CLCA and CLCB in regulating actin patch formation, we performed rescue experiments where either CLC isoforms where overexpressed in CLCA−/− CLCB−/− double knock-out cell lines. Human CLCA or CLCB fused to the red fluorescent protein (RFP) or RFP alone were expressed in both WT and CLCA−/− CLCB−/− U373 cells. Cells were observed using confocal microscopy and analyzed for the presence or absence of actin patches. Fluorescently tagged CLCs have been previously reported to be functional and to be incorporated into clathrin-coated structures upon overexpression into cells [26]. Here, we confirmed that RFP-tagged CLCA and CLCB form discrete foci in U373 cells corresponding to clathrin-coated structures (Figure 2A). The fraction of these clathrin foci also containing AP2 corresponds to CCPs at the plasma membrane (Figure 2A). Importantly, overexpression of either RFP-tagged CLCA or CLCB in WT cells did not alter the actin cytoskeleton compared to mock transfected cells (Appendix A, top panel) or cells expressing RFP alone (Figure 2A, top panel). Quantification did not reveal any significant differences in the number of cells displaying actin patches between WT cells or WT cells overexpressing RFP-tagged CLCA or CLCB (Figure 2B). On the contrary, overexpression of either RFP-tagged CLC isoforms in the CLCA−/− CLCB−/− knocked-out cells lead to a strong reduction of the number of cells displaying actin patches (Figure 2A, bottom panel and Figure 2B, Appendix A, bottom panel and Appendix A). Complementarily, live cell analysis of cells overexpressing the fluorescently-tagged CLCA clearly demonstrates that the disappearance of actin patches coincides with the onset of CLCA expression (Supplementary Movie 1). Together, these results strongly support a model where deletion of CLCA and CLCB induces the formation of actin patches.

Interestingly, a HIP1R binding deficient CLCB mutant (CLCB QQN) has been reported to induce similar actin patch formation [7,16,27] in WT cells. Thus, overexpression of the HIP1R binding deficient mutant in WT cells resulted in the generation of actin patches (Appendix A). Overexpression of CLCB QQN in CLCA−/− CLCB−/− cells was not able to rescue the actin patch formation (Appendix A). Altogether, our results confirm previous observations reporting that deletion of CLCs induce the formation of actin patches at the plasma membrane. Importantly, expression of a single CLC isoform is sufficient to rescue this phenotype highlighting the redundant functions of CLCs in regulating actin patch formation.

### 3.2. CLC Deletion Causes Functional Invadopodia Formation

Deletion of both CLCs induce the formation of actin patches at the plasma membrane (Figure 1C and Figure 2A). Apart from focal adhesions (FA), which are large actin-rich macromolecular structures that are located at the cell periphery and mediate the mechanical linkage of the cell with the extracellular matrix (ECM), invadopodia are another type of actin-containing structures that are located at the ventral plasma membrane mediating interaction with the ECM. Invadopodia are important during embryonic development, tissue positioning, bone remodeling as well as during spreading of metastatic cancer [28,29]. Fluorescence confocal imaging revealed structural similarities between invadopodia and the actin patches observed in cells depleted of CLCA and CLCB (Figure 1 and Figure 2). To confirm whether the actin patches observed upon CLCA and CLCB deletion are invadopodia, WT and CLCA−/− CLCB−/− cells were stained for common invadopodia markers and analyzed using confocal microscopy. We used the actin nucleator Arp3, the matrix metalloproteinase MMP14, zyxin and Cortactin as markers for invadopodia [29,30]. In CLCA−/− CLCB−/− cells, the aberrant actin patches labelled with fluorescent phalloidin were found associated with all invadopodia markers Arp3, MMP14, zyxin and Cortactin (Figure 3A–D). Colocalization analysis using Pearson correlation coefficient revealed that there is significantly more Arp3, MMP14, zyxin and Cortactin associated with actin-rich structures in CLCA−/− CLCB−/− cells compared to WT cells (Figure 3E–H). In WT cells, actin positive structures were only found associated with zyxin (Figure 3C). These structures were identified as FAs as they displayed fibrillar morphology and were located at the periphery of the cells. Interestingly, an enrichment of the clathrin adaptor AP2 in actin patches co-localizing with invadopodia markers was observed in CLC depleted cells (Figure 3A–D and Appendix A) which is in agreement with a previous report showing that clathrin structures and invadopodia are often found in close proximity [31,32]. Together, these results strongly suggest that the actin patches resulting from deletion of CLCs are invadopodia.

An additional hallmark of invadopodia is their protease activity driving local degradation of the ECM during cell migration and invasion [29,33]. To fully demonstrate that our observed actin patches are functional invadopodia, WT and CLCA−/− CLCB−/− cells expressing zyxin fused to the fluorescent protein mCherry (mCherry-zyxin) were seeded on fluorescently labelled gelatin. Using live cell confocal microscopy, we monitored gelatin digestion over time and spatially correlated digestion locations with the position of our actin patches. As expected, in WT cells, digestion of the extracellular gelatin occurred at the leading edge of the cell and was driven by FAs. Digestion of the ECM in the middle of the cell body was only rarely observed in WT cells (Figure 4A–C and Supplementary Video S2). Interestingly, although similar digestions were observed at the leading edge of CLCA−/− CLCB−/− cells, we also observed a pronounced digestion of gelatin directly underneath the cell bodies and this digestion appeared to be mediated by invadopodia structures marked by mCherry-zyxin (Figure 4D–F, arrow heads and Supplementary Video S3). All together, we could show that upon deletion of both isoforms of CLCs, actin patches assemble at the ventral plasma membrane of cells and these structures display all qualitative and functional hallmarks of invadopodia. As such, our results strongly suggest that CLCA and CLCB participate in the regulation of invadopodia formation.

### 3.3. MMP14 Is Upregulated in CLC Depleted Cells

To address the mechanisms by which deletion of CLCA and CLCB induces formation of invadopodia, we turned our attention to the key function of these structures which is their proteolytic activity mediated by matrix metalloproteinases (MMPs). MMP14 is one of the key components of invadopodia and a critical regulator of their formation and function [34]. It was previously reported that the membrane bound MMP14 is able to induce the accumulation of invadopodia-associated proteins which in turn lead to invadopodia formation [35,36]. To test whether deletion of both isoforms of CLCs lead to MMP14 accumulation at the plasma membrane, which in turn would induce the local formation of invadopodia, we analyzed the relative protein level of MMP14 in WT and CLCA−/− CLCB−/− cells. Western blot analysis revealed that cells depleted of both CLCs contain more MMP14 compared to WT cells (Appendix A). This accumulation of MMP14 was confirmed by immunofluorescence analysis of both WT and CLCA−/− CLCB−/− cells (Figure 3B). Accumulation of MMP14 at the plasma membrane in CLCA−/− CLCB−/− cells could be the result of an altered internalization rate of MMP14 from the plasma membrane or an increased trafficking of MMP14 from intracellular storage compartments (or biosynthetic pathway) to the plasma membrane. To discriminate between both of these possibilities, we utilized a microscopy as well as cytometry-based internalization assay in which the kinetics of uptake of surface MMP14 was measured. First, by fluorescently labelling all surface MMP14 (Figure 5A, total surface MMP14), we could show that CLCA−/− CLCB−/− cells have more MMP14 present at their cell surface compared to WT cells (Figure 5B,C). This finding is consistent with the increased amount of invadopodia at the surface of cells depleted of both CLCs (Figure 3B). To directly address whether the accumulation of MMP14 at the surface of CLCA−/− CLCB−/− cells was the result of an impaired internalization of MMP14, the cell surface pool of MMP14 was first saturated with anti-MMP14 antibody on ice. Cells were then transferred at 37 °C for 30 min to let MMP14 be internalized by the cells. The remaining surface bound MMP14 associated antibody was washed off using a low pH buffer and the internalized pool of MMP14 was revealed using a fluorescent secondary antibody (Figure 5A). Quantification revealed that in cells depleted of both CLCs, more MMP14 was found internalized (Figure 5C). However, normalization of the internalized amount of MMP14 to the total amount of MMP14 at the surface of the cells revealed no differences between WT and CLCA−/− CLCB−/− cells (Figure 5D). These results suggest that although CLCA−/− CLCB−/− cells have more MMP14, the internalization rate of this matrix metalloproteinase is not altered compared to WT cells. 

To directly measure the turn-over rate of the surface MMP14 in WT vs. CLCA−/− CLCB−/− cells, we used a cytometry-based internalization assay. The cell surface pool of MMP14 was first evaluated and results confirm its accumulation at the surface of CLCA−/− CLCB−/− cells compared to WT cells (Figure 6B). Following saturation of MMP14 with an anti-MMP14 antibody on ice, MMP14 internalization was followed over time after a temperature shift to 37 °C. The residual anti-MMP14 antibody on the cell surface was detected using a fluorescent secondary antibody (Figure 6A). Flow cytometry analysis of the internalization kinetics of MMP14 normalized to the total amount of surface MMP14 revealed no difference between WT and cell depleted of both CLCs (Figure 6C). Altogether, our results show that accumulation of MMP14 at the plasma membrane of CLCA−/− CLCB−/− knocked-out cells is not due to impaired internalization of MMP14.

### 3.4. Loss of CLCs Leads to Increased MMP14 Intracellular Trafficking towards the Plasma Membrane

To address whether the accumulation of MMP14 was the result of increased MMP14 trafficking towards the plasma membrane, we used Fluorescent Recovery after Photobleaching (FRAP). To focus our analysis strictly on the surface exposed MMP14, we exploited the GFP derived pHluorin protein. This pH sensitive fluorescent protein displays very little to no fluorescence when localized in an acidic environment (e.g., endosomes) while its fluorescence intensity increases in neutral or basic pH [37]. As such, when fused to MMP14 (MMP14-pHluorin), the fraction of MMP14 localized in the intracellular trafficking compartments is not fluorescent while the fraction located at the cell surface is. When expressed in WT cells, MMP14-pHluorin displays a typical uniform membrane localization while in CLCA−/− CLCB−/− cells, MMP14-pHluorin reveals plasma membrane-associated structures that are reminiscent of invadopodia (Figure 7A). Importantly, we found that these structures were positive for markers of invadopodia (e.g., Arp3 and zyxin) and were associated with active digestion of fluorescent gelatin (data not shown). These observations confirm that fusion of MMP14 to pHluorin does not interfere with its incorporation into invadopodia and with their functions. FRAP measurement in WT or CLCA−/− CLCB−/− cells expressing MMP14-pHluorin revealed that following photobleaching, the fluorescence of both membrane-associated and invadopodia-associated MMP14 recovered over time (Figure 7A,B). Analysis revealed that although the recovery rate (T-half: time to recover half the fluorescence intensity) was identical between WT and CLCA−/− CLCB−/− cells (Figure 7C), the fluorescence intensity fully recovered in cells depleted of both CLCs while only 80% of the fluorescent signal was found in WT cells after recovery (Figure 7B,D). These results show that the mobile fraction of MMP14 in CLCA−/− CLCB−/− cells is greater compared to WT cells. This observation strongly suggests that trafficking of MMP14 from intracellular compartments toward the plasma membrane is more efficient in cells depleted of both CLCs. To further confirm this finding, we employed a biotinylation approach using a non cell-permeable biotin analogue to label the proteins and receptors present on the cell surface, in order to dissect the uptake and recycling to the plasma membrane of biotin-labelled surface MMP14 in more detail (Appendix A). To precisely determine the fraction of “internalyzed”, “total” and “non recycled” MMP14, we first controlled the efficiency of the stripping approach by removing the biotin groups from labelled proteins (see methods for details and Supplementary Figure 6C). We then quantified and compared the ratio of labelled endocytosed and recycled (re-trafficked to the cell surface) MMP14 with labelled endocytosed but non-recycled (still internalized) MMP14 from WT and CLCA−/− CLCB−/− cells (Appendix A, schematic and Appendix A). CLCA−/− CLCB−/− cells were able to recycle more MMP14 from the endocytosed biotin labelled MMP14 pool compared to WT cells confirming that CLC deletion causes an upregulation in MMP14 recycling to the plasma membrane (Appendix A).

Complementarily, we controlled the impact of protein neo-synthesis on surface MMP14 levels by blocking protein synthesis using cycloheximide treatment (Appendix A). The observed overall increase in surface MMP14 in both WT and CLCA−/− CLCB−/− over time during cycloheximide treatment is likely due to the fact that the surface pool of MMP14 is constantly internalized and recycled but is tightly controlled and remains relatively stable (Appendix A). The newly synthesized MMP14, however, decreases over time due to the block of protein neosynthesis and has a direct impact on the total amount of MMP14 (Appendix A). Both WT and CLCA−/− CLCB−/− surface/total MMP14 ratios increase equally over time, suggesting that WT and CLCA−/− CLCB−/− cells are equally affected by cycloheximide treatment (Appendix A). Even more strikingly, after 12h of cycloheximide treatment, almost 50% of the total MMP14 pool of CLCA−/− CLCB−/− cells consist of the surface MMP14 fraction, which indicates even further that the increase in MMP14 surface trafficking is due to increased recycling and not due to protein neosynthesis. (Appendix A). Taken together, our results show that deletion of both CLCA and CLCB impairs the intracellular recycling of the matrix metalloproteinase MMP14, promoting its accumulation at the cell plasma membrane.

### 3.5. Loss of MMP14 Prevents Invadopodia Formation in CLCA−/− CLCB−/− Cells

Finally, to demonstrate that MMP14 accumulation at the plasma membrane of CLCA−/− CLCB−/− cells is directly responsible for invadopodia formation [38], we knocked-down MMP14 in WT and CLCs depleted cells using shRNA (Figure 8A). While in CLCA−/− CLCB−/− cells, most of the cells (>80%) display invadopodia structures, upon silencing of MMP14, we observed a significant reduction of the number of cells having invadopodia (Figure 8B,C). Importantly, knock-down of MMP14 did not impact the organization of the fibrillar actin cytoskeleton (stress fibers) (Figure 8B). Interestingly, knock-down of MMP14 in WT cells also resulted in the depletion of the few invadopodia structures that are naturally present in these cells (Figure 8B,C).

All together, our results show that deletion of both CLCA and CLCB promotes MMP14 accumulation on the cell surface which in turn induces the formation of functional Invadopodia structures. These findings highlight a novel function of CLCA and CLCB in regulating invadopodia formation.

## 4. Discussion

Deletion of both CLCs in cells causes the assembly of abnormal actin patches. In this work, we identify the nature of these actin structures and could demonstrate that they show all characteristics of invadopodia rather than disorganized actin-cytoskeleton patches. Identification of these actin patches as functional invadopodia was based on the presence of common invadopodia markers as well as spatially increased proteolytic activity (gelatine digestion). Formation of these invadopodia structures requires the loss of both CLCA and CLCB but could be rescued by overexpression of either CLCs, suggesting that both CLCA and CLCB have a redundant function in regulating their formation. Finally, we found that formation of these structures was due to the accumulation of MMP14 at the plasma membrane in CLCA and CLCB depleted cells. This accumulation was not due to a defect in CME and MMP14 turn over from the plasma membrane but rather due to an increased intracellular trafficking of MMP14 towards the plasma membrane. Furthermore, we also showed that knock-down of MMP14 reduced invadopodia formation in CLC depleted cells, demonstrating that MMP14 is key for invadopodia formation upon CLC deletion. Thus, our work further highlights the fact that CLCs have limited functions in regulating endocytosis but play a critical role in regulating intracellular trafficking and sorting of cargo to the plasma membrane.

Deletion of both CLCs induced actin patch formation at the ventral plasma membrane (Figure 1). These observations confirm the long known interplay between CLCs and the actin cytoskeleton as similar actin accumulations at the plasma membrane were previously reported upon siRNA mediated knockdown of CLCs [7,21,22]. Accumulation of these actin structures at the plasma membrane required the deletion of both CLCA and CLCB. Individual knock-out of CLCA or CLCB was found to be not sufficient for inducing the formation of these invadopodia structures and this actin patch phenotype could be rescued by expression of either one of the CLC isoforms (Figure 2), highlighting the redundant function of both CLCs in regulating actin patch/invadopodia formation. These findings are in agreement with previous works [7,21]; however, it was shown that in the HeLa cell line, transient knock down of CLCA alone was sufficient for inducing the formation of actin structures at the plasma membrane [23]. While we cannot exclude that these discrepancies might reflect cell type specific different functions of CLCA and CLCB, it is possible that these discrepancies arise from the different strategies used to deplete CLCs: knock-out (in this work) vs. siRNA mediated knock-down CLCs [23]. In cells transiently depleted of one CLC, the remaining CLC does not have the time to take over the impaired function of the depleted one. On the contrary, in cells knock-out for one CLC, the remaining CLC will be able to functionally replace the depleted one due to the time necessary to perform the single cell clonal expansion of the knock-out cell line. We actually observe a higher expression level of CLCB in cells knocked-out for CLCA, and vice versa (Figure 1A). Within this possibility, this would suggest that although both CLCs can compensate for each other loss, in physiological condition, CLCA would be responsible for regulating actin patch formation. However, a recent study reported decreasing levels of MMP14 upon CLCB silencing [39] and we observe the same finding in decrease in MMP14 expression in CLCB−/− cells (Appendix A), suggesting that CLCB is also involved in regulating invadopodia formation.

Overexpression in cells of a CLCB mutant lacking the binding site of the actin organizer HIP1R [16] (Appendix A) and siRNA-mediated knock-down of HIP1R also induced the formation of similar actin patches [7,21,40,41]. As HIP1R was shown to interact with cortactin at endocytic sites and regulate actin filament formation [42] it was proposed that these aberrant actin structures upon HIP1R deletion/depletion are due to impaired actin filament formation through loss of HIP1R binding to cortactin [7,40]. Indeed, we observe an accumulation of cortactin at actin structures in cells (Figure 3D) as well as an attenuation of HIP1R in clathrin-coated structures in CLC depleted cells (Appendix A). Here we described that the actin structures that accumulate upon deletion of CLCA and CLCB are functional invadopodia [7,43,44]. These invadopodia structures contain integrins, focal adhesion-related proteins (i.e., talin, zyxin), actin and actin polymerization and branching mediating proteins (Cortactin, WASP, Arp2/3). Interestingly, we also observed the presence of AP2 in these invadopodia structures (Figure 3 and Appendix A), and recruitment of AP2 seems to take place after digestion of the extracellular environment by MMP14 (Figure 4E,F). In some rare instances, AP2 forms unusual filamentous “tail-like” structures which appear to follow actin (Figure 3A). The origin and function of these structures remain to be determined, but we hypothesize that clathrin structures are involved in the activity of invadopodia, probably by mediating local recycling of membranes and internalization of extracellular compounds. This is consistent with the observation that clathrin structures are found at invadopodia [31,32].

We could show that upon CLC deletion, the trans-membrane metalloproteinase MMP14, a key component of invadopodia and essential for ECM degradation [41], was upregulated at the protein level (Appendix A) and was enriched at the ventral plasma membrane (Figure 3B, Figure 5B and Figure 6B). The reasons for MMP14 accumulation into cells are not clear but are very likely associated with its accumulation at the plasma membrane in invadopodia structures. MMP14 surface expression is tightly regulated and upregulation of MMP14 coincides with malignant cancer progression [45,46,47]. Interestingly, a recent publication demonstrated that the MMP14 interaction with the ECM alone, independently of the catalytic activity is important for assembly of mature invadopodia [36]. Thus, the described elevated MMP14 surface levels of CLC knock-out cells (Figure 3B, Figure 5B and Figure 6B) alone might be sufficient to prime invadopodia formation, since silencing of MMP14 in CLCA−/− CLCB−/− cells abolished invadopodia formation (Figure 8). Given the importance of CME to regulate the surface expression of MMP14 [34], we monitored the kinetic of MMP14 uptake from the plasma membrane using either flow-cytometry or confocal microscopy (Figure 5 and Figure 6). We could show that the endocytic rate of MMP14 was not significantly different between WT and cells depleted of CLCA and CLCB (Figure 5C and Figure 6C). Thus, altered endocytosis of MMP14 in CLCs depleted cells is likely not the reason for MMP14 surface accumulation.

Using a FRAP microscopy approach, we were able to show that the surface accumulation of MMP14 was the result of an increased trafficking of MMP14 from intracellular compartments to the plasma membrane (Figure 7). The half-time of fluorescence was the same between WT and CLCA−/− CLCB−/− cells, indicating similar intracellular trafficking rates (Figure 7D). Interestingly, we could show that while WT cells only recovered to about 80% of their initial fluorescence, cells depleted of both CLCs fully recovered, indicating that deletion of CLCA and CLCB increases the mobile fraction of MMP14 (Figure 7B,C). Recently, it was shown that knockdown of the Phosphatidylinositol 4-kinase IIβ (PI4KIIβ) similarly induces the formation of invadopodia [35]. This was caused by increased trafficking of MMP14 via exocytic endosomal compartments to the plasma membrane leading to invadosome formation. As such, it is possible that PI4P levels and CLCs co-operate to regulate MMP14 trafficking to the plasma membrane. Upon CLC deletion, trafficking of several cargo was reported to be perturbed: CIMPR was trapped in the TGN and recycling of β1 integrin as well as signaling receptors like TGFβR2, CXCR4, and DOR were impaired [7,9,21]. As such, we propose that CLCs negatively regulate MMP14 to prevent MMP14 accumulation and invadopodia formation.

Interestingly, the actin binding protein cortactin was proposed to not only be involved in forming the actin scaffold for invadopodia but rather promote the secretion of secretory vesicles containing MMPs including MMP14 [46,48]. Together with the previously described mechanisms that HIP1R binding to cortactin inhibits actin assemblies at endocytic sites, and with the fact that CLC deletion disrupts the localization of HIP1R to endocytic sites [7,42], we propose the following model: Deletion of CLCA and CLCB causes upregulation of MMP14 recycling to the plasma membrane (Figure 9). The consequence of CLC deletion is the disruption of HIP1R recruitment to endocytic sites, more specifically to cortactin. In turn, the unbound cortactin drives actin polymerization, which causes actin patch formation. Cortactin simultaneously promotes secretory vesicles containing MMP14, thereby causing accumulation of both actin and MMP14. The actin structures as well as MMP14 clustering form the scaffold for further invadopodia maturation.

## Figures and Tables

**Figure 1 cells-10-00451-f001:**
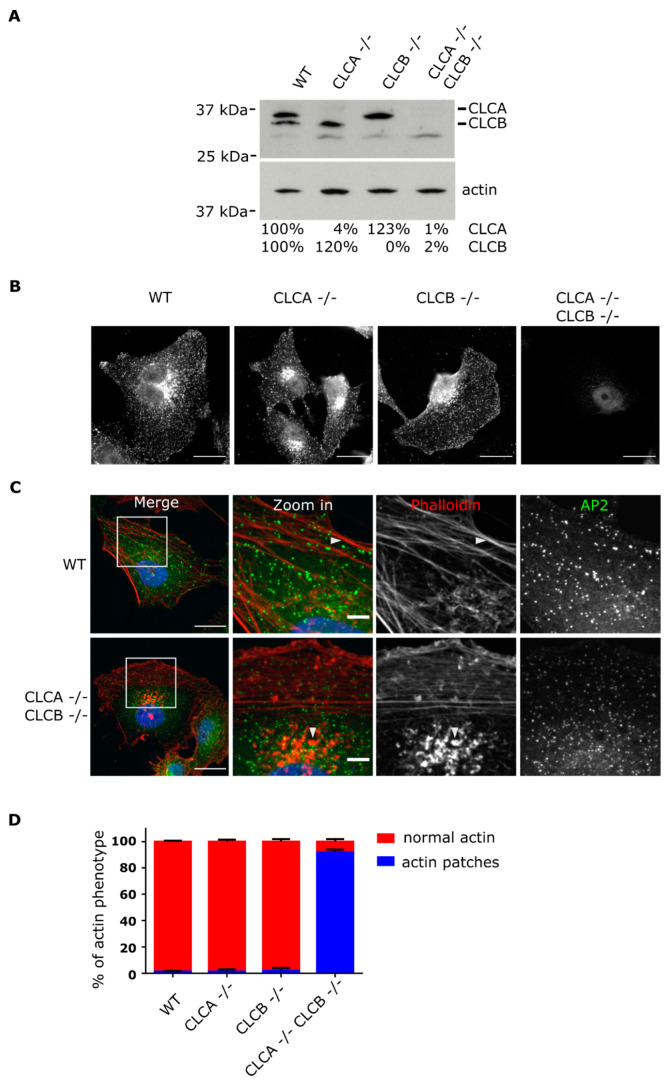
Knockout of CLCA and CLCB induces formation of actin patches at the ventral plasma membrane. (**A**) Western blot analysis of WT, CLCA−/−, CLCB−/− and CLCA−/− CLCB−/− cell lines using an antibody against CLC and actin. (**B**) Representative epifluorescence images of genome edited U373 cells. CLCs were immunostained using an anti-CLC antibody, scale bar indicates 20 µm. (**C**) Representative confocal images of WT, CLCA−/−, CLCB−/− and CLCA−/− CLCB−/− cell U373 cells stably expressing the clathrin adaptor AP2 fused to GFP (green). Actin was stained with phalloidin Alexa Fluor 647 (red). Zoom in of actin structures, scale bar indicates 20 µm for merge and 5µm zoom in. White arrows indicate representative actin structures. (**D**) Quantification of relative percentage of actin phenotype among U373 WT, CLCA−/−, CLCB−/− and CLCA−/− CLCB−/− cell lines. Cells displaying actin patches were quantified, displayed as the mean and SD, n = 3, 300 cells per condition.

**Figure 2 cells-10-00451-f002:**
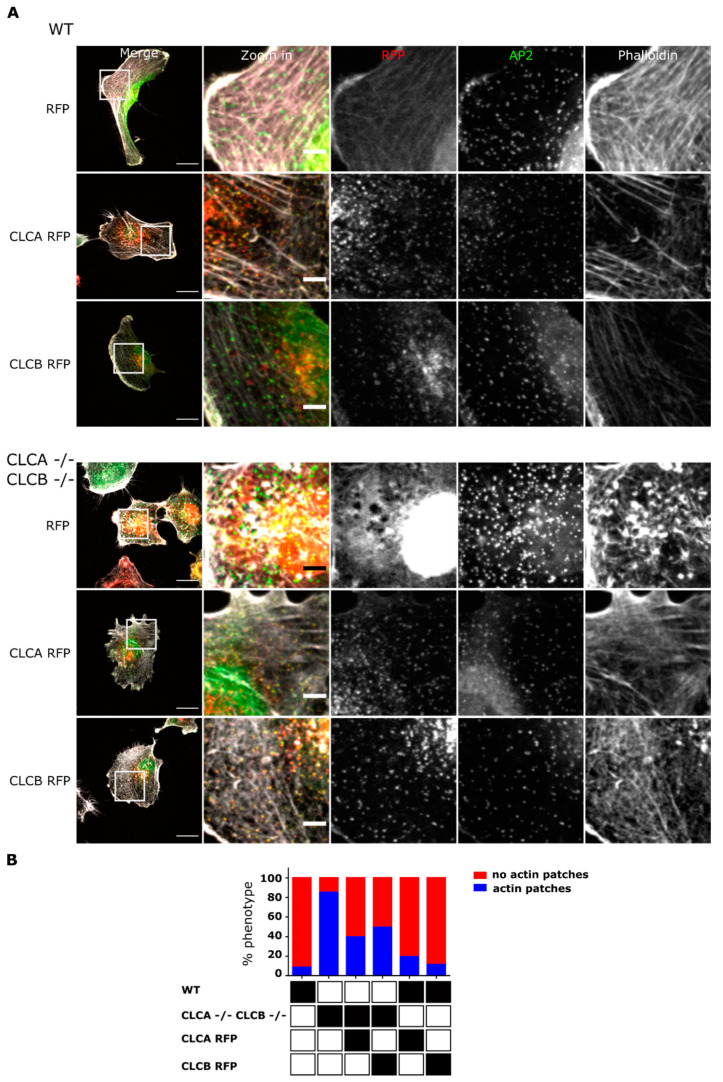
Trans-complementation of CLCs in CLCA−/− CLCB−/− cells rescues actin patch formation. (**A**) U373 WT and CLCA−/− CLCB−/− cells stably expressing the clathrin adaptor AP2 fused to GFP (green) were transfected with either RFP tag alone, CLCA RFP or CLCB RFP (red). Cells were fixed and stained with Alexa fluor 647-labelled phalloidin for actin visualization (grey). Representative confocal images are shown, scale bar indicates 20 µm for merge and 5µm zoom in. (**B**) Relative percentage of actin phenotype among cell population. Cells displaying actin patches vs. cells without actin patches were quantified. Only RFP-positives were counted and used for analysis. 10 fields of view were analyzed. WT and RFP 161 cells, WT and CLCA RFP 133 cells, WT and CLCB RFP 92 cells, CLCA−/− CLCB−/− and RFP cells 175 cells, CLCA−/− CLCB−/− and CLCA RFP 184 cells, CLCA−/− CLCB−/− and CLCB RFP 210 cells.

**Figure 3 cells-10-00451-f003:**
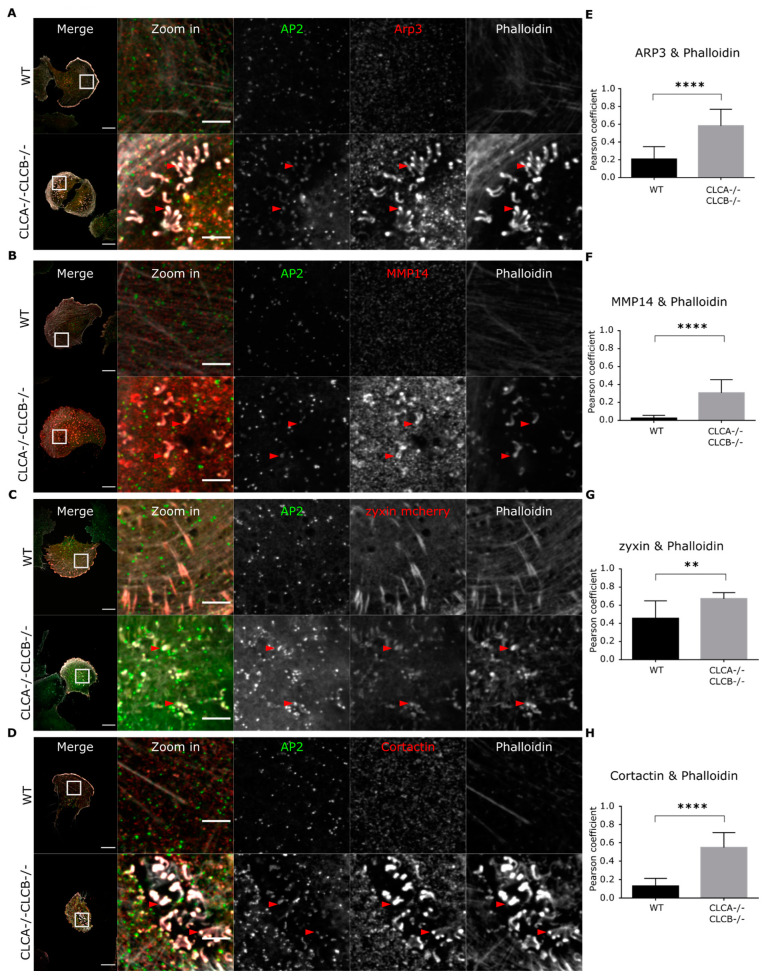
Actin patches in CLCA−/− CLC−/− cells co-localize with common invadopodia markers. U373 AP2-GFP WT and CLTA−/− CLTB −/− cells were fixed 24 h post seeding. Cells were stained with an antibody against (**A**) Arp3 (red) (**B**) MMP14 (**D**) Cortactin (red) or transduced with (**C**) BacMam expressing mCherry-zyxin (red). Cells were analyzed by confocal microscopy for co-localization with AP2 (green) and actin (grey). Representative confocal images are shown. The scale bar represents 20 µm for the merge and 5 µm for the zoom region, respectively. Red arrows indicate co-localizing structures. (**E**–**H**) Pearson correlation analysis. Shown are the mean with SD. Statistical analysis: unpaired *t*-test. ARP3 8 fields of view, **** *p* < 0.0001; MMP14 13 fields of view, **** *p* < 0.0001; zyxin, 12 fields of view, ** *p* = 0.0014; Cortactin 11 fields of view, **** *p* < 0.0001.

**Figure 4 cells-10-00451-f004:**
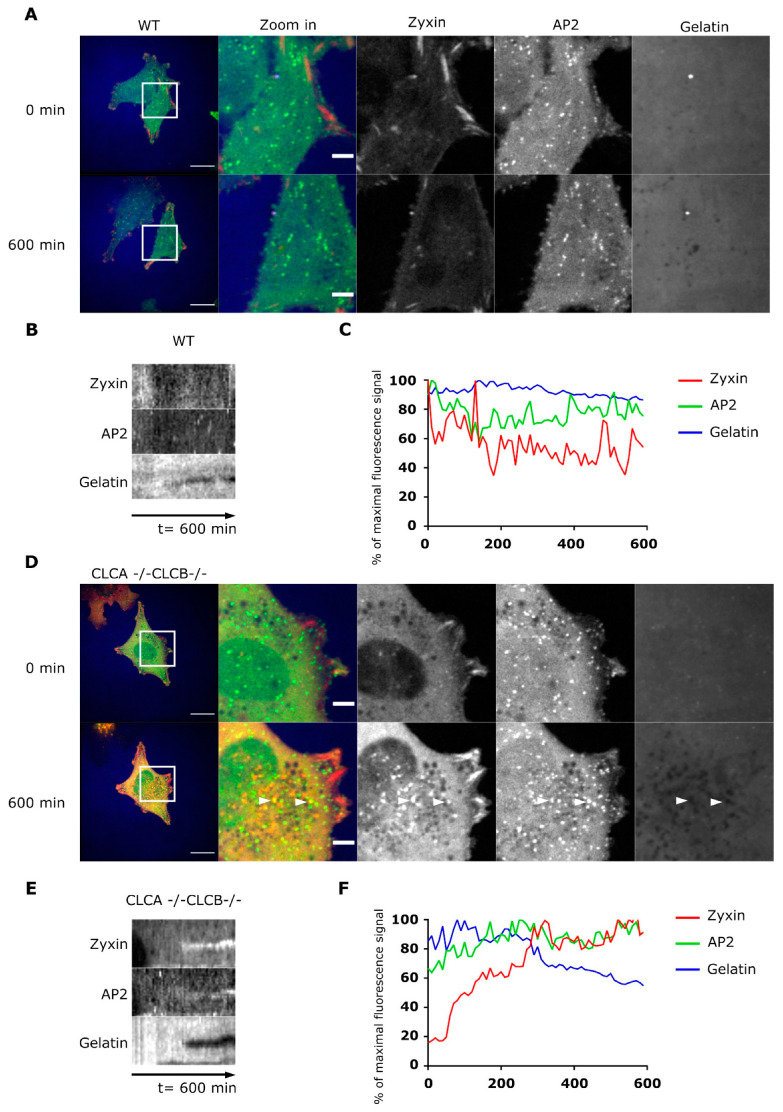
Actin patches in CLCA−/− CLCB−/− cells display proteolytic activities. (**A**) Representative live-cell confocal spinning disc microscopy of U373 WT cells stably expressing AP2-GFP (green) and transiently expressing mCherry-zyxin (red) seeded onto Alexa Fluor 647-labelled gelatin (blue) coated coverslips. Live-cell confocal imaging was performed for 600 min. Here, representative pictures at the beginning and at the end of imaging are displayed. The scale bar equals 20 µm and 5 µm for the “full cell” view and zoom in region, respectively. (**B**) Kymograph of WT U373 cells expressing AP2-GFP and mCherry-zyxin seeded on fluorescent gelatin (600 min). (**C**) Representative fluorescence intensity profiles overtime of WT U373 cells expressing AP2-GFP (green), mCherry-zyxin (red) seeded on Alexa Fluor 647-labelled gelatin (blue). (**D**–**F**) same as (**A**–**C**) except for CLCA−/− CLCB−/− U373 cells.

**Figure 5 cells-10-00451-f005:**
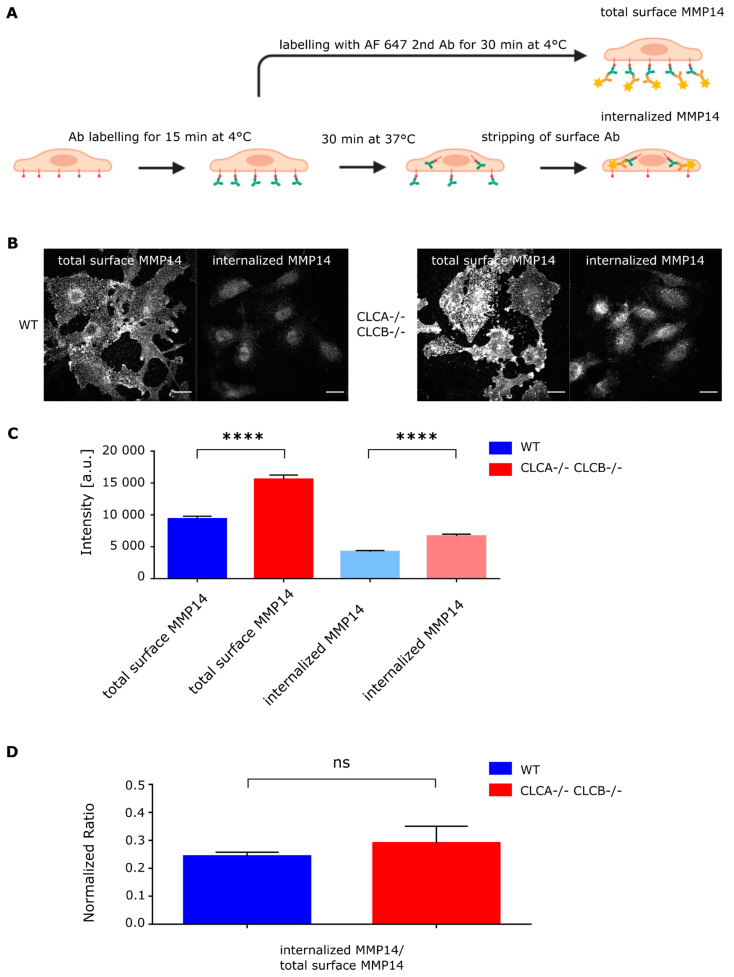
Knock out of CLCA and CLCB does not impair MMP14 internalization. (**A**) Schematic of microscopy based internalization assay. Cells plated on glass coverslips overnight were serum starved for 1 h, labelled with MMP14 antibody on ice for 15 min. For total surface MMP14 levels cells were directly fixed after primary antibody incubation and stained using the Alexa Fluor 647 labelled anti-rabbit secondary antibody (without permeabilization). For internal MMP14 levels cells were incubated at 37 °C for 30 min in order to start MMP14 internalization. For internal MMP14, cell surface was stripped using ice-cold PBS at pH 2.5, followed by fixation and immunostaining with permeabilization. Imaging was performed using confocal microscopy. Image sections were acquired every 0.5 µm for a total of 7.5 µm. Fiji was used to quantify MMP14 levels. (**B**) Representative confocal images for total surface and internal MMP14 fractions of U373 AP2-GFP. (**C**) Quantification of MMP14 signal intensities. Shown are the mean with SD. WT total surface 37 cells; WT internalized 38 cells; CLCA−/− CLCB−/− total surface 49 cells; CLCA−/− CLCB−/− internalized 51 cells. Shown are the mean with SD. Statistical analysis: unpaired *t*-test: total surface MMP14, **** *p* < 0.0001; internalized MMP14, **** *p* < 0.0001. (**D**) Intensity ratios from (**C**) of internalized MMP14/total surface MMP14 in U373 AP2-GFP WT and CLCA−/− CLCB−/− cells. Shown are the mean with SD. Statistical analysis: unpaired *t*-test: not significant (ns).

**Figure 6 cells-10-00451-f006:**
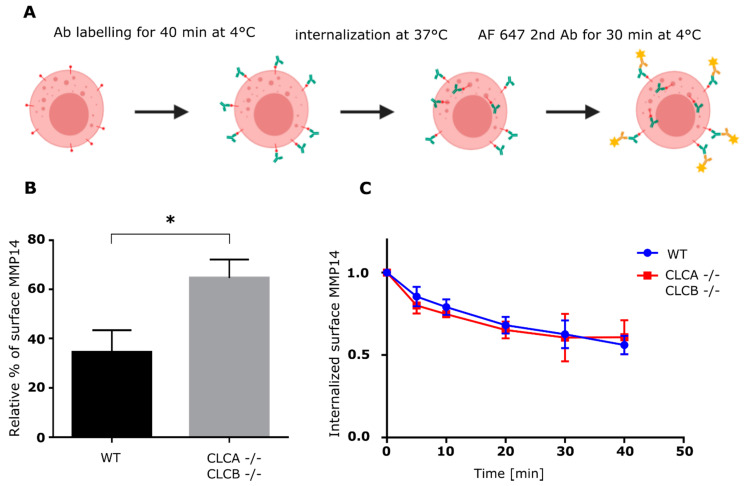
MMP14 expression and internalization in U373 WT and CLCA−/−CLCB−/− cells. (**A**) Schematic of flow cytometry internalization assay. Cells were dissociated from cell culture flasks resuspended in serum free culture medium and kept on ice before incubation with saturating concentration of MMP14 antibody. Internalization was initiated by incubation at 37 °C for the respective time intervals between 0 and 40 min. After MMP14 internalization, cells were incubated with anti-rabbit Alexa Fluor 647 (1:10000) for 30min, washed and analyzed by flow cytometry (FACS Canto II, BD Biosciences). Data were analyzed using FlowJoTM version 10B). (**B**) Surface MMP14 was normalized to total MMP14. Relative percentage was calculated from 3 independent experiments. Median with SD is shown. Statistical analysis: unpaired t test,* *p* = 0.0111. (**C**) Flow cytometry-based MMP14 internalization assay. Shown are relative MMP14 surface levels normalized to 0 min. Each point represents mean and SD of three biological replicates.

**Figure 7 cells-10-00451-f007:**
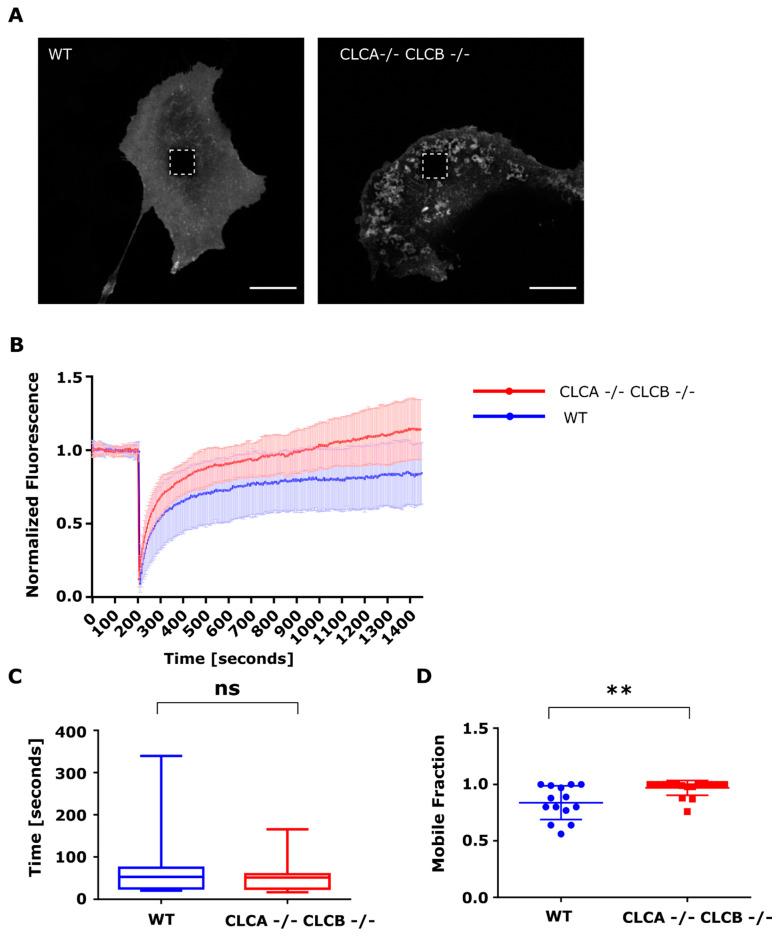
Fluorescence recovery after photobleaching of MMP14-pHluorin. (**A**) Representative live-cell confocal spinning disc microscopy of U373 WT and CLCA−/− CLCB−/− transiently expressing MMP14-pHluorin. Live-cell confocal imaging was performed for a total of 25 min at a frame rate of 5 s per frame. The scale bar equals 20 µm. Highlighted area indicates FRAP region. (**B**) Normalized fluorescence of FRAP recovery curves. Normalized mean fluorescence intensity profiles of WT (blue) and CLCA−/− CLCB−/− cells (red) before and after FRAP. Shown are mean and SD computed from 14 WT and 18 CLCA−/− CLCB−/− cells FRAP recovery curves. Standard deviations are shown. (**C**) t-Half of MMP14-pHluorin in U373 WT and CLCA−/− CLCB−/− cells. T-Half was calculated from 14 (WT) and 18 (CLCA−/− CLCB−/−) FRAP recovery events, each FRAP event was monitored in a different cell. Mean with SD are shown. Statistical analysis: unpaired t test, ns. All FRAP analysis was performed using the EasyFRAP-web tool. (**D**) Mobile fraction of MMP14-pHluorin in U373 WT and CLCA−/− CLCB−/− cells. Mobile fraction was calculated from 14 (WT) and 18 (CLCA−/− CLCB−/−) FRAP recovery events. Mean with SD are shown. Statistical analysis: unpaired t test, ** *p* = 0.0069. (**D**) t-Half of MMP14-pHluorin in U373 WT and CLCA−/− CLCB−/− cells. T-Half was calculated from 14 (WT) and 18 (CLCA−/− CLCB−/−) FRAP recovery events, each FRAP event was monitored in a different cell. Mean with SD are shown. Statistical analysis: unpaired *t* test, not significant (ns). All FRAP analysis was performed using the EasyFRAP-web tool.

**Figure 8 cells-10-00451-f008:**
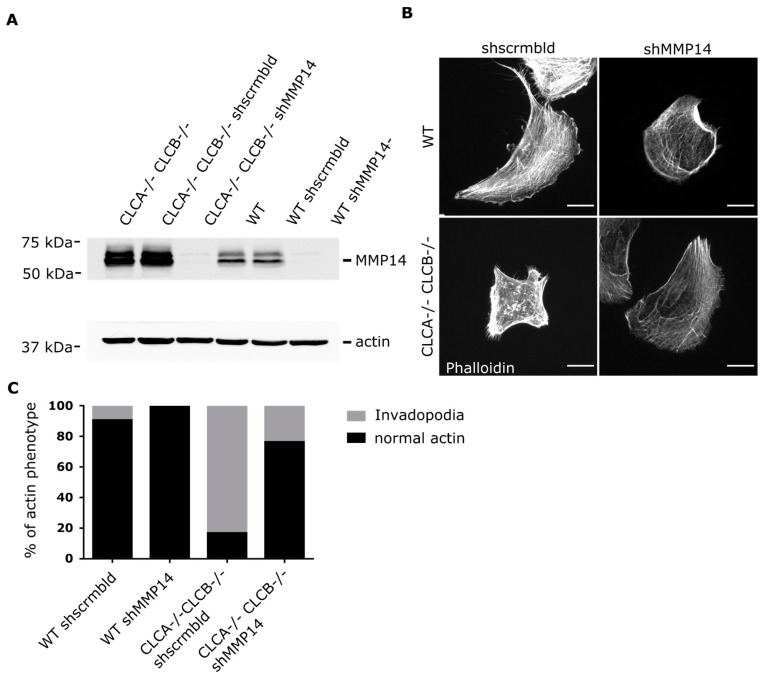
MMP14 attenuation in CLCA−/− CLCB−/− cells prevents invadopodia formation. (**A**) Western blot analysis of WT and CLCA−/− CLCB−/− cell lines showing the protein levels of cells stably expressing Mock, scrambled shRNA or an MMP14 shRNA construct. Actin protein levels were used as control. (**B**) Representative widefield microscopy images of WT and CLCA−/− CLCB−/− cells stably expressing scrambled shRNA or an MMP14 shRNA construct, scale bar indicates 20 µm. (**C**) Quantification of relative percentage of actin phenotype among U373 WT and CLCA−/− CLCB−/− cells stably expressing scrambled shRNA or an MMP14 shRNA construct. 10 fields of view were analyzed. WT shscrmbld 23 cells, WT shMMP14 13 cells, CLCA−/− CLCB−/− shscrmbld 25 cells, CLCA−/− CLCB−/− shMMP14 21 cells.

**Figure 9 cells-10-00451-f009:**
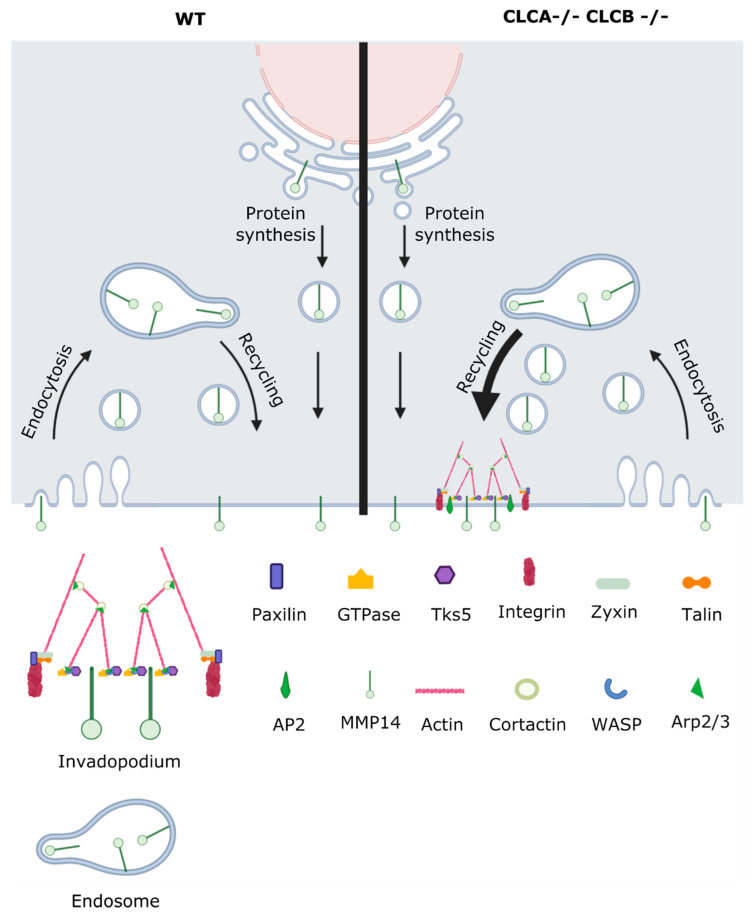
Loss of CLCs induces invadopodia formation by actin polymerization and MMP14 recruitment. Schematic showing upregulation of MMP14 recycling in CLCA−/− CLCB−/− cells causing invadopodia formation.

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
