# Peer review of "Role of Clathrin Light Chains in Regulating Invadopodia Formation"

_cells, 2021, doi:10.3390/cells10020451_

Round 1

Reviewer 1 Report

In their manuscript “ Role of clathrin light chains in regulating invadopodia formation”, Mukenhirn and co-workers address the role of different clathrin light chain isoforms for intracellular trafficking. They use CRISPR/Cas9-mediated genome editing to generate CLCA  and CLCB deletion cell lines and then study the impact of these deletions.

Notably, they find that cells lacking either CLCA or CLCB show only minor defects, suggesting that these two isoforms hold redundant functions. In contrast, their analysis of the double deletion shows defects in the actin organization. Specifically, they find that double deletion cell lines show the accumulation of actin patches, which can be rescued by expression of tagged CLCA or CLCB.

The authors then continue to characterize these structures in detail and propose that they are invadopodia , based on morphology and the presence of molecular markers. They further bolster their data by measuring degradation of fluorescent gelatin.

This finding is then followed by an investigation how these structures are formed. The authors find that the matrix metalloproteinase MMP14 – which is a key component of invadopodia - is enriched on the surface of CLCA/CLCB knockout cells. Based on the functions of clathrin in endocytosis, they first investigate if CLCA/CLCB knockout results in altered endocytosis of MMP14. They find that endocytosis is not altered, but rather that more MMP14 is transported to the plasma membrane. Finally, the authors then test if the increased invadopodia formed in CLCA/CLCB knockout cells are dependent on MMP14. To this end, they use shRNA to deplete MMP14 in both wild-type and double knockout cells. Using this system, they find that the appearance of invadopodia in CLCA/CLCB KO cells is dependent on MMP14.

The data presented in this paper is clear, and the authors use several different approaches to investigate their hypotheses. The performed experiments are well-controlled and the presented results are convincing.

The main missing piece in the puzzle is where the MMP14 pool that drives the enhanced secretion in CLCA/B KO cells originates and  - more importantly - how CLCA/CLCB regulates this process. The specific role of CLCA/B remains unexplored.

The authors propose that CLCA/CLCB acts at endocytic sites via the actin-binding protein HIP1R, which in turn negatively regulates cortactin. Loss of CLCA/B leads to an increase of cortactin-dependent actin polymerization and secretion of MMP14-carrying vesicles, which in turn - by means of a feedback loop – stabilize the formation of stable actin structures and thus invadopodia.

However, this reviewer sees several problems with this hypothesis, which is not strongly supported by the data. First, the authors go a long way to show that the observed structures are functional invadopodia – does this imply that most invadopodia are formed by at endocytic sites? This is not really consistent with the presented data -  CLCA/CLCB is present at multiple endocytic sites (as shown in Figure 1B) and yet the invadopodia are primarily formed in proximity of the nucleus, and there is only limited overlap of AP2-labelled endocytic sites and the putative sites of invadopodia formation. Moreover, the authors show (in the Kymograph in Figure 4E) that AP2 arrives after gelatin degradation, further arguing against this simplistic model.

In addition, they show that the invadopodia is dependent on MMP14, which would argue against the relatively simple model that the pure loss of HIP1R recruitment is enough to drive actin polymerization – in this case one would expect that endocytic sites should show strong actin polymerization even in the absence of MMP14.

While this reviewer sees that an in-depth investigation of CLCA/CLCB function is beyond the scope of this manuscript, the proposed model and discussion on the potential function is mostly speculative, and a deeper and more carefully worded discussion -including a model figure - would be desirable.

Specific comments:

When referring to knockouts, the authors use “depletion” -  “deletion” would be more appropriate.

Figure 1A:

The authors report weak residual expression (between 1-4 %) – this could be either contamination of their cell line with a low level of WT cells or the generation of aberrant mRNAs in the Kos. Especially Clone CLCA -/- shows a weak band slightly at ~37 kDA which is absent in both the WT and the other KO clones. Were the KO clones characterized on the genomic level? Please include sequences of the individual used clones as supplemental data.

Is the increased expression of CLCA in CLCB KOs (and vice versa) an artefact specific for the shown western or are the cells compensating?

Figure 3:

A: AP2 in the CLCA/B Ko panel shows tail-like structures similar to actin and ARP3– is this cross-excitation/ bleedthrough or a real localization? The bright ARP3 / actin signals have a high risk of bleedthrough , please include a proper control

A-D: It would be informative – especially for the proposed model – to measure AP2 colocalization with the observed structures. Since AP2 spots are small, Pearsons will unlikely give satisfactory results, so “Manders” colocalization might be an appropriate way to test this.

Figure 7:

  • FRAP might not be the best way to measure trafficking, since transmembrane proteins such as MMP14 diffuse fast and most of the bleached pool will be replenished from the surrounding areas. It should be possible to measure the rate of exocytosed MMP14 using the pHluorin-MMP14, e.g. after bleaching the cell surface and then counting secretion events.
  • The higher mobility of MMP14 in KO cells is not fully in line with the authors model where MMP14 is part of a platform which induces more invadopodia formation – should in this case the mobility not be lower? Also, why do the authors see an increase in the FRAP over the base-line? This is noteworthy, since one would assume that normally the plasma membrane pool should be relatively stable.
  • Where does the “additional” MMP14 originate? Is it secretion of newly-synthesized MMP14 or is there more recycling?

General comments:

  • It might be preferable to show the cell biological data as individual data points instead of barplots or boxplots, e.g. as “Superplots” and use the corresponding statistics on a “per experiment” base – see the discussion of Lord et al, JCB 2020 (https://pubmed.ncbi.nlm.nih.gov/32346721/ )

Reviewer 2 Report

The manuscript from Boulant and colleagues explores further the role of clathrin light chains (CLCS) in membrane trafficking. Using CRISPr engineered cells, lacking both CLCA and CLCB, the authors demonstrate that the actin rich puncta previously identified when CLC function is disrupted correspond to invadipodia. Moreover they show that these structures apparently occur because of an accumulation of MMP14, a key driver of invadopodia, at the cell surface. Using two different approaches, they demonstrate that the loss of the CLCs does not account for the increased cell surface expression of MMP and FRAP experiments demonstrated an enhanced mobility of MMP14 in the absence of CLCs.

This is a nicely written and clearly presented manuscript, which extends our understanding of the interaction of CLCs with actin. In particular it is of interest that the disorganized actin structures are now identified as invadipodia, structures that are relevant in both normal and patho-physiology. It is also of interest that the authors have shown roles for CLC in modulating intracellular trafficking as well as internalization from the cell surface. The presumed mechanism and location of where the CLCs act intracellularly is not however addressed by the authors and this undermines the manuscript as it stands. Either CLCs are involved in the regulated recycling of MMP14 from early endosomes, thus extending the life time of a population of MMP14, or they are involved in regulated delivery of newly synthesized MMP14 from the secretory pathway. If the former, this would be the opposite of the reported role of CLCs in the recycling of integrins. In my view the manuscript would be considerably strengthened by the inclusion of a couple of straightforward experiments to distinguish between these two possibilities:

  1. The authors could directly measure recycling of MMP14 in wild-type and CLC negative cells using a biotinylation protocol.
  2. The authors could explore whether the increase in cell surface expression in CLC negative cells is maintained following treatment with cyclohexamide.

Round 2

Reviewer 1 Report

The authors have adressed all my concerns, and the newly added model figure nicely supplements the manuscript.